# CardiCat: a Variational Autoencoder for High-Cardinality Tabular Data

## Abstract

High-cardinality categorical features are a common characteristic of mixed-type tabular datasets. Existing generative model architectures struggle to learn the complexities of such data at scale, primarily due to the difficulty of parameterizing the categorical features. In this paper, we present a general variational autoencoder model, CardiCat, that can accurately fit imbalanced high-cardinality and heterogeneous tabular data. Our method substitutes one-hot encoding with regularized dual encoder-decoder embedding layers, which are jointly learned. This approach enables us to use embeddings that depend also on the other covariates, leading to a compact and homogenized parameterization of categorical features. Our model employs a considerably smaller trainable parameter space than competing methods, enabling learning at a large scale. CardiCat generates high-quality synthetic data that better represent high-cardinality and imbalanced features compared to competing VAE models for multiple real and simulated datasets.

## 1 Introduction

Recent years have seen dramatic improvement in the generative modeling of complicated stimulus including images, audio and most recently natural languages. Generative models characterize the joint distribution of the variables, and allow sampling from this distribution. They can therefore be used for imputing missing values, regenerating data while hiding sensitive information, and detecting anomalies. Variational Auto Encoders (VAEs, (Kingma & Welling, 2013)) are generative models composed of two neural-networks: the *encoder* transforms a data example into a distribution on the $p$-dimensional latent space, and the *decoder* transforms a vector sampled from the latent space into a data example. The sampling increases the smoothness of the decoder, and the result is often more interpretable and better reflects the diversity of the original data.

Generative models and VAEs in particular are most successful when the data they are trained on is homogeneous, meaning that the individual features are similarly distributed, and there are local structures governing the interaction between features (e.g. pixels or words) (Ma et al., 2020; Suzuki & Matsuo, 2022). In contrast, generative models still struggle when modeling one of the most common types of datasets, the tabular data (Nazabal et al., 2020). In tabular datasets, each example can be comprised of a set of data-fields from different types (*mixed types*), including numerical, integer, and character strings. In addition, the features can exhibit complex dependencies, but the order or topology is often arbitrary and provides little information regarding this structure (in contrast to homogeneous signals such as images or language). Prominent examples include electronic medical records (EMR), personal credit default, e-commerce and behavioral datasets from social networks.

High cardinality categorical features is another pervasive characteristic of mixed-type tabular data. The cardinality of some categorical features, meaning the number of possible unique values, can be extremely high, with severe imbalances between the values (Xu & Veeramachaneni, 2018). Such high-cardinality categorical features are often very informative - consider the information on a patient contained in the in features such as "diagnosis", "occupation" or "city / state". For a generative model to learn the behavior of a high-cardinality feature, the interplay between different values of the feature need to be identified. Often, the decoders prefer not to *guess* rare values, and this leads to a collapse in the marginal distributions towards the common categories. More technically, current models usually require categorical features to be initially parameterized by one-hot encoding or string similarity encoding such as in Cerda & Varoquaux (2020), which relies on the information

between pairs of categories. Yet, one-hot encoding high-cardinality features have many detrimental effects on neural networks (Rodríguez et al., 2018; Avanzi et al., 2024). One-hot encoding forces a flat label space (where each value is equally different), which might disregard the complex relationships that exist amongst the values within a categorical variable. One-hot encoding also dramatically increases the dimension of inner layers of the network, leading to poor statistical properties, memory constraints and a strain on the optimization process (Choong & Lee, 2017).

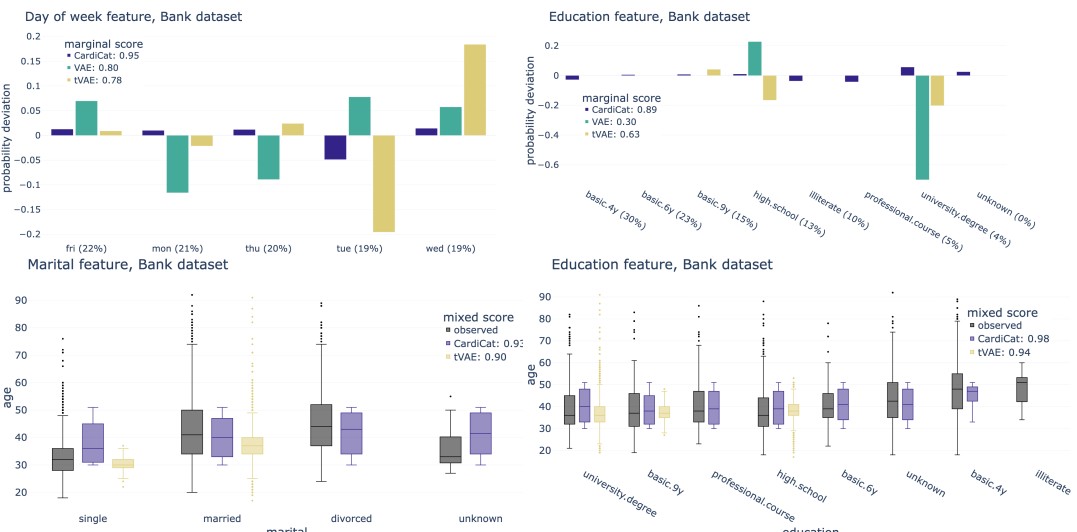

Figure 1: Top: bar plots show deviation between generated categorical probabilities from three models (Cardi-Cat in blue, VAE in green, tVAE in yellow) and true probabilities on three features. Shorter bars mean better reconstruction. TV score is in the legend. Bottom: the box plots show the conditional distribution of a numerical variable given a categorical one. The true distributions are in black, and the generated ones are in blue (CardiCat) and yellow (tVAE). Missing box plots means that categorical value was not sampled. Overall CardiCat better reconstructs the marginal and bi-variate distributions.

We propose a VAE-based framework for the modeling and synthesis of tabular data; one that can efficiently represent high-cardinality categorical features in the context of mixed feature types. Our framework, CardiCat, adds embedding layers for the high-cardinality categorical features. The learned embedding layers are low-dimensional numeric vector representations of the values of the categorical features. CardiCat's embedding layers are efficiently learned in tandem as part of the complete encoder-decoder network optimization. Therefore, the topology of the embedding layer can be influenced by the the joint-distribution of the categorical features and the other features in the data. A key innovation is that the decoded output (and likelihood) of these features is evaluated in the smooth and homogenized embedding space rather than in the observed categorical space. This allows us to avoid all together the need to one-hot encode the non-binary categorical features at any point in the process, thus reducing the number of trainable parameters significantly, and feeding the network more homogenized data. As we show in our experiments, this encoding allows the VAE to better recover the marginal and joint distributions of categorical features (see Figure 1).

Our contributions can be summarized as follows: (1) We propose a method to introduce regularized categorical embedding into VAEs in a way that efficiently homogenizes categorical features and prevents embedding layers collapse. We show that our method's capacity to represent and accurately reconstruct the marginal and multivariate trends of the data surpasses that of previous models. In addition, it requires considerably fewer learned parameters; (2) We develop a public benchmarking framework for tabular VAEs composed of simulated and real-life tabular datasets emphasizing high-cardinality categorical features; And (3) We make available an open-source implementation of the regularized embeddings architecture that can easily be adapted to different VAE frameworks.

The use of embedding layers to represent categorical features in neural networks is not new. However, to our knowledge, this is the first public and benchmarked end-to-end tabular generative model that uses dual-architecture regularized embedding layers for parameterizing categorical features.

## 2 VARIATIONAL AUTOENCODERS FOR TABULAR DATA

This section provides a brief overview of VAEs and their shortcomings as they relate to heterogeneous data. It then discusses the main strategies for accommodating diverse data types.

### 2.1 VARIATIONAL AUTOENCODERS

Variational autoencoders are deep latent-variable generative models that are based on autoencoders (AE) (Kingma & Welling, 2013). Latent-variable models assume that each data-point $\mathbf{x} \in \mathbf{X}$ is generated from a true, latent distribution $p_\theta(\mathbf{z})$ of some unknown random real-valued vector $\mathbf{z}$ of dimension $a$. VAEs are different from autoencoders in their ability to learn the generating distribution of the data. In contrast to AE, the VAE encoder outputs a set of parameters that defines the latent distribution. In addition, VAEs impose regularization constraints on the latent distribution. This forces the latent distribution to be as close as possible to a predefined prior distribution. Each random value of the latent vector $\mathbf{z}$ that is fed through the decoder should produce a meaningful output. Variational Bayesian inference is used to derive a tractable lower-bound on the likelihood of the data, which can be maximized by gradient optimization:

$$\log p_\theta(\mathbf{x}) \geq E_{\mathbf{z}}[\log p_\theta(\mathbf{x}|\mathbf{z})] - D_{kl}(q_\phi(\mathbf{z}|\mathbf{x})|p_\theta(\mathbf{z}))$$

The right hand side is called the evidence lower bound (ELBO) of the log likelihood of the data $\log p_\theta(\mathbf{x})$. The ELBO is comprised of two terms, the reconstruction term $E_{\mathbf{z}}[\log p_\theta(\mathbf{x}|\mathbf{z})]$ and the KL-divergence term $D_{kl}(q_\phi(\mathbf{z}|\mathbf{x})|p_\theta(\mathbf{z}))$ that acts as a regularizer. The reconstruction term describes how well the generated output resembles the input data.

### 2.2 ADAPTIONS OF VAEs FOR TABULAR DATA

In mixed-type tabular data, the features of each datapoint include both categorical and numerical features. Modeling mixed tabular data is often difficult for VAEs due to the heterogeneous mixed-type nature of the features. The model must be flexible enough to incorporate simultaneous learning of discrete and numerical features with different distribution characteristics.

In adapting VAEs to heterogeneous tabular data, we can identify several approaches:

**Accommodating prior:** When learning VAEs for complex and high-dimensional data, the standard-normal VAE prior is often (1) too strong and over-regularizes the encoder, and (2) not expressive enough to represent well the underlying structure of the data. This can lead to the problem of posterior collapse: when the encoder fails to distill useful information from $x$ into the variational parameters of the posterior. In addition, the standard normal distribution assumes data points are centered around zero with a standard deviation of one, which is not a good fit for mixed-type tabular data. The use of more flexible priors, such as Gaussian mixture model (GMM) priors with components that are learned through back-propagation, can aid in the avoidance of over-regularization (Tomczak & Welling, 2018; Guo et al., 2020; Apellániz et al., 2024).

**Type-specific likelihoods:** This approach (implemented in RVAE and HI-VAE (Akrami et al., 2020; Nazabal et al., 2020)) deals with the heterogeneity by providing each feature type (e.g. real-valued, categorical) with a different likelihood model. For example, on categorical features the decoder network may output a vector of probabilities and the loss would be based on the cross-entropy score. Note that in recent empirical studies HI-VAE has failed to recover the observed marginal distributions as well as failed to surpass competing methods (Ma et al., 2020; Gong et al., 2021).

**Type homogenization:** An alternative approach is to transform all input variables into Gaussian variables to produce a homogeneous VAE. In VAEM(Ma et al., 2020), the authors propose a two-stage structure that avoids using type-specific likelihoods altogether. In the first stage individual features are homogenized by learning an independent VAE for each feature. Then, the separately learned factorized latent variables from the first stage models are used as inputs to a second stage VAE. However, projecting categorical features into real-values without relying on other covariates may result in subpar results (see Section 4). Focusing only on numerical data, Xu et al. (2019) tVAE applies a mode-specific normalization using a variational Gaussian-mixture model (VGM) for each numerical feature.

**A conditional generator** model is used to generate samples that are conditioned on some specific values of the data or its associated labels. This is done by adding the condition to the encoder and the decoder's input so it can be better represented in the latent space and provide more control over the generated output (Sohn et al., 2015). Conditional generators for VAEs are often used to add categorical label information (as the conditioned label) to the homogeneous latent space (Mishra et al., 2018; Lim et al., 2018). For tabular data with multiple discrete features, there are multiple ways to set up the conditional generators.

All these extensions and adaptations strive to alleviate the VAE's sensitivity to fail under complex, multi-dimensional, non-normal data. However, they do not provide a sufficient solution for dealing with high-cardinality heterogeneous tabular data. High-cardinality features exacerbate the heterogeneity of the input data. This is due to the need to one-hot encode features as long binary vectors. Furthermore, one-hot inflates the space of the network's trainable parameters, which places a heavy burden on the computational resources needed to train the model. Finally, because high-cardinality features are likely to be imbalanced, their minor values bear relatively small effect on the network's optimization regime, resulting in vanishing estimated marginal probability. Therefore, improving and adapting tabular VAEs to work with high-cardinality heterogeneous data can have a drastic effect on their synthetic data generation abilities. In this work we focus on type homogenization and propose an embedding-based VAE framework that aims to homogenize both the categorical features and the likelihood models.

## 3 THE CARDICAT MODEL

Our model deals with non-binary categorical features by embedding them into a low dimensional space using information from the entire encoder-decoder neural network. CardiCat deviates away from a traditional VAE by (a) substituting one-hot categorical encodings with embedding presentations throughout the network and its loss, and (b) by adding a loss regularization term to prevent embedding-collapse. These learned embeddings can be thought of as smoothed-out categorical parameterizations that are learned from the network's reconstruction loss. This allows us to efficiently parameterize discrete features in a self-learned mechanism that depends on the embedding space and not on the original features. Our framework is implemented on a simple VAE architecture in order to cleanly demonstrate the advantages of such framework.

### 3.1 NOTATIONS FOR CATEGORICAL FEATURES AND EMBEDDINGS

Consider a mixed-type tabular dataset $\mathbf{D} = \{\mathbf{x}\}_{i=1,..,n}$, where each of the $n$ datapoints $\mathbf{x}_i = (x_{i,1}, \ldots, x_{i,m})$ is a vector of $m$ features, some numerical and some categorical. We denote the j'th feature vector $\mathbf{x}_j$, and $\mathrm{x}_j$ when we consider it a random variable. Let $H \subseteq \{1, ..., m\} = [m]$ be the set of *categorical features*, marking any discrete feature with more than two values. For simplicity, the domain of a categorical feature $\mathrm{x}_j$ with *cardinality* $c_j$ is identified with $[c_j]$. We denote the categorical distribution of the $\mathrm{x}_j$ by $Cat(\pi_1, ..., \pi_{c_j})$, meaning that $P(\mathrm{x}_j = \ell) = \pi_\ell$ for $\ell \in [c_j]$.

We equip each categorical feature $\mathrm{x}_j$ with an embedding, a learned mapping from a categorical value to a real-valued vector $emb_j : [c_j] \to R^{k_j}$. The embedding dimension $k_j$ is often chosen such that $k_j << c_j$. The embedding can be represented using matrix $\mathbf{e}_j = (e_{j,1}, ..., e_{j,c_j})'$. $\mathcal{E}$ represents the set of all embeddings parameters $\mathcal{E} = \{\mathbf{e}_j; j \in H\}$.

### 3.2 CARDICAT'S DUAL ENCODER-DECODER EMBEDDINGS

In a traditional neural network with embedding layers, the embeddings are learned from back-propagating the loss gradients throughout the network. For example, in supervised learning, where each sample consists of a feature vector and target label pair $(\mathbf{x}_i, y_i)$, the embedding layers are learned using the network's back-propagation gradients from $\mathcal{L}(\hat{y}_i, y_i)$. The resulting embedding geometry codes the relationship between different categorical values and the target.

Generative model architectures such as VAE require a different solution for integrating unsupervised embeddings. Because the network's decoder is tasked with generating the encoder's inputs, the embedding layers must be also represented as part of the decoder's output. Therefore, CardiCat employs dual encoder-decoder embedding layers architecture where the embeddings appear both as

trainable layers in the encoder, and in the output of the decoder where they actively participate in calculating the loss (instead of the original features). At each propagation step, the decoder tries to reconstruct the embedding vectors, and any deviations from the expected embeddings will be penalized by the ELBO. The reduced dimension space is therefore learned by back-propagating the gradients of the ELBO through both the decoder's and the encoder's embedding weights.

### 3.3 LOSS & EMBEDDING REGULARIZATION

CardiCat is composed of (a) embedding layers $\mathcal{E}$ that (b) feed into an encoder network $q_\phi(\mathbf{x}) = (\mu_\phi(\mathbf{x}), \sigma_\phi(\mathbf{x}))$ defining the mean vector and coordinate-wise variance of the latent $p$-dimensional Gaussian vector, and (c) a decoder network $p_\theta(\mathbf{z})$. We optimize the network using the following loss based on the weighted variational lower bound (ELBO):

$$\mathcal{L}_{\phi,\theta,\mathcal{E}}(\mathbf{x}) = \mathcal{L}_{Recon}(\hat{\mathbf{x}}, \mathbf{x}; \mathcal{E}) + \lambda_1 \cdot D_{KL}(N(\mu_\phi(\mathbf{x}), \sigma_\phi(\mathbf{x})I) \parallel N_p(0, I)) + \lambda_2 \cdot Reg(\mathcal{E}).$$

The main novelty compared to a simple VAE is that the reconstruction loss of the categorical features is computed in the embedding space. The decoded output of the VAE for a categorical value $x_j$ is a $k_j$ dimensional numerical vector $\hat{\mathbf{e}}_j = p_{j,\theta}((z)) \in R^{k_j}$, and it is compared to the embedding vector of the true feature $\mathbf{e}_j(\mathbf{x}_j)$:

$$\mathcal{L}_{Recon,j}(\hat{\mathbf{x}}, \mathbf{x}; \mathcal{E}) = ||\mathbf{e}_j(x_j) - \hat{\mathbf{e}}_j||^2.$$

For binary or numerical features, we use the standard cross-entropy and mean-squared error loss. Because the embedding weights are homogenized and dense numerical presentations, they are treated the same as the other numerical features, and there is no need to type-separate the conditional likelihood between these features. The full reconstruction loss is the sum over all features: $\mathcal{L}_{Recon}(\hat{\mathbf{x}}, \mathbf{x}; \mathcal{E}) = \sum_j \mathcal{L}_{Recon,j}(\hat{\mathbf{x}}, \mathbf{x}; \mathcal{E})$.

We add a regularization term on the embedding weights to prevent an embedding collapse. Because the embeddings are learned in tandem (encoder & decoder), there is a risk that embedding vectors become too similar, artificially decreasing the loss. The embedding regularization penalizes changes to the total coordinate-wise variance of the embedding, compared to the initialization:

$$V_j(\{\mathbf{e}_j\}) = \frac{1}{k_j} \sum_{\ell=1}^{k_j} var(e_{j,\ell}), \qquad Reg(\mathcal{E}) = \frac{1}{|H|} \sum_{j \in H} (V_j(\{\mathbf{e}_j\}) - V_j(\{\mathbf{e}_j^0\}))^2$$

Once the loss gradients are calculated, the embedding layers are adjusted as part of the back-propagation step.

### 3.4 MODEL ARCHITECTURE

Figure 2 provides an overview of the CardiCat architecture. The input of the encoder is separated into data types: binary features $x_l, l \in L$, categorical features $x_h, h \in H$, and numerical features $x_m, m \in M$. Binary features are one-hot encoded, while categorical features ($c_j \geq 3$) are encoded to a reduced embedding space of size $k_j$ (hyper-parameter). Using the parameterization trick, the encoder $q_\varphi(\mathbf{z}_i|\mathbf{x}_i)$ outputs $\mathbf{z}_i \sim N_a(\boldsymbol{\mu}, diag(\boldsymbol{\sigma})) \in \mathbb{R}^a$. The decoder outputs data types identical to the encoder's input so that loss can be calculated by assuming a factorized Gaussian likelihood model $p(\mathbf{x}_i|\mathbf{z}_i) = \prod_j p_j(x_{i,j}|\mathbf{z}_i)$[1].

This suggested architecture was chosen for the following reasons. First, the VAE embeddings provide a natural homogenized parameterization of otherwise difficult to learn parameterizations such as one-hot encoding. Second, the construction of each embedding space depends on the entirety of data, promoting the further sharing of information throughout the VAE network. This is due to the fact that the presentations are learned and depend on the network-wide optimization and loss function. In addition, the geometries of the learned embedding spaces can reveal by analyzing the distance between different values of each category. Finally, avoiding one-hot encoding and instead using end-to-end embedding layers provides a meaningful reduction in the number of parameters needed to train tabular VAEs. In the next chapter we show that these adaptations offer significant advantages over other VAE architectures for learning high-cardinality large-scale tabular data.

---

[1]A full description of the network structure and layers can be found in the supplementary

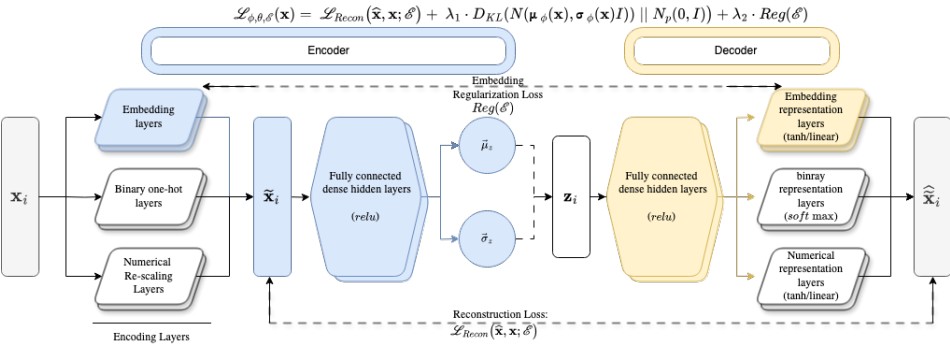

Figure 2: Illustration and description of CardiCat's network architecture.

## 3.5 CONDITIONAL EMBEDDING GENERATOR

We also provide a conditional generator variant of CardiCat, where it is easier to generate synthetic data with specific categorical properties. To be able to represent the conditional embedding vector as an additional input to the encoder and the decoder's networks, CardiCat first adds a "mask" value to each of the categorical features before training. The conditional *embedding* vector $< \mathbf{e}_1^{mask}, \mathbf{e}_2^{mask}, \dots, \mathbf{e}_j, \dots >$ is composed of the concatenation of all the embedded masked values $\mathbf{e}^{mask}$ for non-conditional features, and the embedded values $\mathbf{e}_j$ if $x_j$ belong to the set of conditional variables. This conditional embedding vector inherits its values from the encoder-decoder's embedding layers, but its weights are non-trainable during training.

## 4 EXPERIMENTS

In this section we evaluate and benchmark CardiCat against competing approaches and baselines. We measure the quality of the generative model, meaning how well the distribution of the generated synthetic data matches that of the original sample, specifically on the categorical features.

Table 1: Benchmark datasets and total trainable parameters

| dataset | datapoints | features | total cardinal | total params VAE | total params CardiCat |
|---|---|---|---|---|---|
| PetFinder | 11,537 | 14 | 193 | 122k | 78k |
| Bank | 32,950 | 21 | 53 | 86k | 79k |
| Census | 32,561 | 15 | 102 | 98k | 79k |
| Medical | 188,806 | 8 | 1,146 | 226k | 75k |
| Credit | 210,201 | 17 | 121 | 102k | 81k |
| Criteo | 406,654 | 11 | 1,724 | 359k | 85k |
| MIMIC | 556,617 | 11 | 693 | 249k | 82k |
| Simulated | 100,000 | 11 | 87 | 97k | 78k |

## 4.1 BENCHMARK MODELS, DATASETS & SETUP:

**Models.** We compare our proposed model `CardiCat` against three different models: `VAE`, `tVAE`, and `tGAN`. Acting as a baseline, `VAE` is a vanilla VAE model with a standard-normal latent structure, one-hot encoded categorical features, and standardized numerical features. `tVAE` adds a mode-specific normalization using a variational Gaussian-mixture model (VGM). In addition, we use `tGAN` as a comparison against a GAN with a conditional generator architecture. Both `tVAE` and `tGAN` are as specified in (Xu et al., 2019). All models are defined with hyper-parameters and network structure that resemble those of the `VAE` model as closely as possible (size and depth of hidden layers, epochs, optimizer, etc.). `VAEM` was not included in our evaluation results due to unsupported published code and subpar performance of our implementation of the model[2].

**Datasets**. Seven real-world datasets and one simulated dataset are used to benchmark the competing models (Table 1). The datasets were selected for their high-cardinality features and complex joint-

---

[2]More information on the models can be found in the supplementary

distributions. PetFinder, Bank, Credit, and Census are commonly used machine learning mixed tabular datasets from the UCI machine learning repository (Dua & Graff, 2017) or from Kaggle. MIMIC (MIMIC-III) is a dataset of intensive care medical records (Johnson et al., 2016). Medical is a Medicare dataset that provides information on the use, payment, and hospital charges of more than 3,000 U.S. hospitals. Criteo is a large-scale online advertising dataset for millions of display ads and users, including properties, signals and behavior. The different datasets vary in size, cardinality, and distributional complexity. In addition, a simulated dataset (simulated) was generated by sampling dependent and independent pairs of categorical, numerical and mixed features. [3]

**Experimental setup**. Each dataset was split 80/20 into disjoint $train$ and $test$ subsets. The train dataset was used for fitting the generative model. Then, synthetic data from each trained model was generated by sampling from the model's prior distribution in the latent space, and feeding these latent samples into the model's decoder. Lastly we decode the output to de-normalize numerical features and map categorical features back to their original labels. Details including model specification, hyper-parameters and data processing can be found in the supplementary materials. We then compare the statistical properties of these generated data samples to those of the test subset.

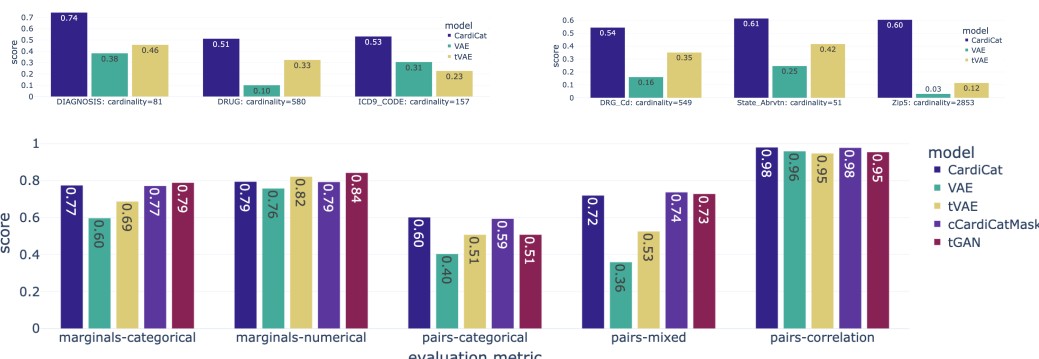

Figure 3: Top: Marginal reconstruction TV scores of high-cardinality features (left MIMIC, right Medical). Higher is better. Bottom: Average evaluation results by feature or feature-pair type. Scores represent (from left) marginal scores for categorical and numerical features, and then scores for categorical, mixed and numerical pairs; scores are averaged across all relevant features or feature pairs, and across datasets. The first three bars represent VAE models (CardiCat, VAE, tVAE), and the last two represent conditional generators.

## 4.2 EVALUATION METRICS

We compare the generated synthetic data against the $test$ sample of each original dataset. For each dataset, we evaluate both the marginal and the bi-variate distribution reconstruction of each feature and pairs of features. Exact formulations are found in the supplementary materials.

**Marginal reconstruction**. Continuous features were evaluated using the complement of the Kolmogorov-Smirnov (KS) statistic. Categorical features were evaluated using the complement of the Total Variation Distance (TVD).

**Bi-variate reconstruction** Numerical feature pairs were evaluated by taking the complement of the correlation difference. Categorical feature pairs were evaluated by taking the complement of the TVD on the contingency table. Mixed feature pairs were evaluated by looking at the conditional distributions (numerical given each value of the categorical). A KS statistic was measured for each value between observed and generated samples. These KS statistics were combined using a weighted average, and their complement was recorded.

## 4.3 RESULTS

We compare the ability of the different models to recreate the marginal distributions and the joint distributions of numerical, categorical and mixed-type variables. Table 2 summarizes the average

---

[3]More information on the simulated and the other datasets can be found in the supplementary.

benchmark results of three runs for each model and dataset. Figure 3 summarizes the evaluation metrics over all datasets.

**Data reconstruction**. `CardiCat` outperforms the other VAE models in marginal and bi-variate reconstruction of categorical features, while still competitively and accurately reconstructing numerical marginals and bi-variate distributions.

Table 2: Evaluation results of VAEs for all datasets. The results average across all relevant feature or feature pairs in dataset, for three different training runs of the generative model. Standard deviations across training runs are averaged per metric and shown in the parenthesis. Best results are in bold.

| dataset | model | marginal | | pairs | | |
| | | categorical (0.01) | numerical (0.01) | categorical (0.01) | mixed (0.045) | correlation (0.015) |
|---|---|---|---|---|---|---|
| Bank | CardiCat | **0.86** | 0.78 | **0.77** | **0.72** | **0.97** |
| | VAE | 0.71 | 0.70 | 0.52 | 0.44 | 0.96 |
| | tVAE | 0.76 | **0.81** | 0.59 | 0.58 | 0.94 |
| Census | CardiCat | **0.82** | **0.75** | **0.70** | **0.67** | **0.99** |
| | VAE | 0.81 | 0.73 | 0.68 | 0.42 | 0.98 |
| | tVAE | 0.82 | 0.75 | 0.68 | 0.64 | 0.95 |
| Credit | CardiCat | **0.93** | **0.83** | 0.94 | **0.79** | **0.98** |
| | VAE | 0.78 | 0.62 | 0.82 | 0.38 | 0.86 |
| | tVAE | 0.91 | 0.77 | **0.97** | 0.68 | 0.93 |
| Criteo | CardiCat | **0.71** | **0.82** | **0.51** | **0.63** | **0.97** |
| | VAE | 0.55 | 0.65 | 0.29 | 0.34 | 0.95 |
| | tVAE | 0.62 | 0.69 | 0.36 | 0.46 | 0.95 |
| MIMIC | CardiCat | **0.80** | 0.87 | **0.66** | **0.80** | **1.00** |
| | VAE | 0.68 | 0.74 | 0.45 | 0.75 | 0.98 |
| | tVAE | 0.77 | **0.90** | 0.58 | 0.51 | 0.94 |
| Medical | CardiCat | **0.59** | 0.82 | **0.16** | **0.58** | 0.96 |
| | VAE | 0.13 | 0.71 | 0.01 | 0.06 | 0.96 |
| | tVAE | 0.29 | **0.88** | 0.06 | 0.23 | **0.97** |
| PetFinder | CardiCat | **0.87** | 0.76 | **0.77** | **0.72** | **0.99** |
| | VAE | 0.75 | 0.76 | 0.57 | 0.41 | 0.97 |
| | tVAE | 0.83 | **0.77** | 0.70 | 0.54 | 0.98 |
| Simulated | CardiCat | **0.77** | **0.84** | **0.63** | **0.78** | **1.00** |
| | VAE | 0.56 | 0.84 | 0.32 | 0.38 | 0.99 |
| | tVAE | 0.67 | 0.75 | 0.46 | 0.52 | 0.95 |

**Network's trainable parameters**. Table 1 summarizes also the total number of trainable parameters for different VAE models in our benchmarks. As the complexity, cardinally and the number of categorical features increases in the data, CardiCat advantages in parameter efficiency increases.

**Evaluation against a conditional generator**. Figure 3 also includes the evaluations of `cCardiCatMask` and `tGAN`. `cCardiCatMask` outperforms or is comparable to `tGAN` in most cases, even while `tGAN` employs training-by-sampling in an additional attempt to over come the imbalance training data. Training `tGAN` takes significantly more time than `CardiCat`, for example, orders of magnitude longer in the case of Criteo.

## 5 DISCUSSION

This work attempts to bridge the gap that currently exists with learning high-cardinality tabular data using variational autoencoders. While this type of data is becoming more prevalent, current mixed-type tabular VAE models fail to adequately model and learn high-cardinality features. The CardiCat architecture we propose homogenizes the high-cardinality categorical features through their embedding parameterization. The regularized embeddings are learned as part of the VAE training. From our benchmarks, we show that our model performs significantly better and is able to produce high quality synthetic data compared to other VAE models of comparable size. Our implementation is open-source, and can easily be extended by others.

We note the following limitations of this work. First, the network architecture we use is basic, and neither architecture nor training parameters have been optimized for individual datasets. Though this experimental design choice is deliberate, there is a chance that gains observed here would not carry over to much more sophisticated architectures. Nevertheless, the ideas and code can be easily adapted to additional scenarios. For example, combining a refined model such as the mixture modeling of tVAE should further improve the recovery of numerical features. Second, we evaluate the joint distribution recovery by looking at marginal and pair interactions; we leave downstream effects on supervised learning and interpretation of the learned embeddings for later work.

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
