# Supplementary Material:
# CardiCat: a Variational Autoencoder for High-Cardinality Tabular Data

## 1 Architecture

CardiCat adapts a Variational Autoencoder (VAE) architecture to add regularized dual encoder-decoder embedding layers to parameterize categorical features 1. In contrast to other neural embedding architectures, such as in natural language processing and entity embeddings, CardiCat's embeddings are learned in tandem by the recognition model (encoder) and the generator model (decoder). This architecture dynamically parameterizes and homogenizes the high-cardinality features during training, which accommodates better learning overall.

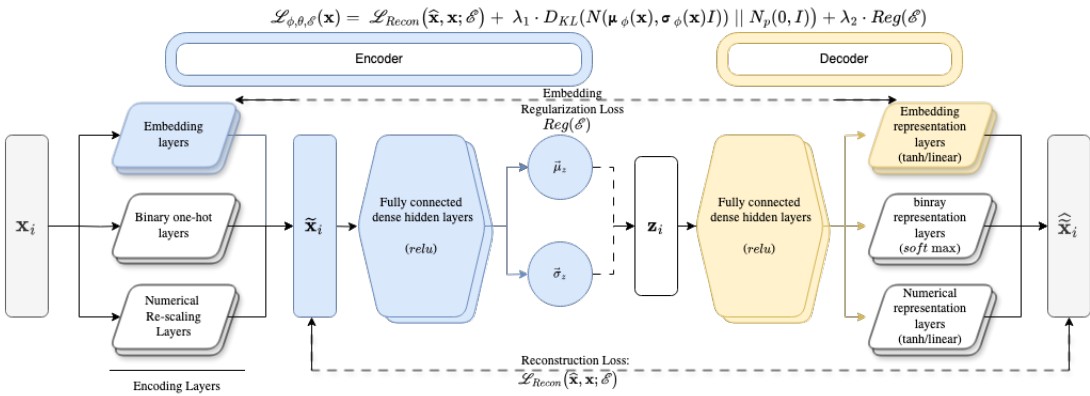

Figure 1: illustration and description of CardiCat's network architecture.

CardiCat's encoder-decoder architecture can be formally describe as:

$$\text{Encoder:} \begin{cases} \mathbf{e}_h = emb(x_h), \quad h \in H \\ \mathbf{c}_l = oh(x_l), \quad l \in L \\ r_m = standard_{1 \to 1}(x_m), \quad m \in M \\ h_1 = reLU(FC_{128 \to 128}(cnct(\mathbf{e}, \mathbf{c}, \mathbf{r}))) \\ h_2 = reLU(FC_{128 \to 128}(h_1)) \\ \boldsymbol{\mu} = FC_{128 \to a}(h2) \\ \boldsymbol{\sigma} = exp(0.5(FC_{128 \to a}(h2)) \\ q_\varphi(\mathbf{z}|[\mathbf{e}, \mathbf{c}, \mathbf{r}]) \sim N_a(\boldsymbol{\mu}, diag(\boldsymbol{\sigma})) \end{cases}$$

$$\text{Decoder:} \begin{cases} h_1 = reLU(FC_{128 \to 128}(\mathbf{z})) \\ h_2 = reLU(FC_{128 \to 128}(h_1)) \\ \bar{\mathbf{e}}_h, \bar{e}_{h,k} = tanh(FC_{128 \to 1}(h2)), \quad h \in H \\ \bar{r}_m = tanh(FC_{128 \to 1}(h2)), \quad m \in M \\ \bar{\mathbf{c}}_l \sim softmax(FC_{128 \to c_l})(h2), \quad l \in L \\ p_\theta([\mathbf{e}, \mathbf{c}, \mathbf{r}]|\mathbf{z}) = \prod_{l=1}^{L} \mathcal{P}(\bar{\mathbf{c}}_l = \mathbf{c}_l) \times \\ \quad \prod_{h=1}^{H} \phi_K(\hat{e}_h = e_h)) \times \prod_{m=1}^{M} \phi_1(\hat{r}_m = r_m) \end{cases}$$

CardiCat is trained with ELBO loss as defined in the main paper. We use the Adam optimizer with learning rate of 0.0005.

## 2 DATASETS DETAILS

More information on the benchmark datasets is available online 1. Identifier and index columns were removed from all datasets (see source code).

Table 1: Sources of benchmark datasets

| Dataset | Source |
|---|---|
| PetFinder | https://www.kaggle.com/competitions/petfinder-adoption-prediction |
| Bank (bank marketing) | https://archive.ics.uci.edu/ml/datasets/Bank%2BMarketing |
| Census (census income) | https://archive-beta.ics.uci.edu/dataset/20/census+income |
| Medical (medicare impatient hospitals) | https://data.cms.gov/provider-summary-by-type-of-service/medicare-inpatient-hospitals/ |
| Credit (home credit default risk) | https://www.kaggle.com/competitions/home-credit-default-risk/data |
| Criteo | http://labs.criteo.com/downloads/2014-kaggle-display-advertising-challenge-dataset |
| MIMIC-III | https://mimic.mit.edu/docs/iii/ |
| Simulated | included in source-code |

## 3 EXPERIMENTS

### 3.1 SOURCE CODE

The source code to our model and benchmarks is available here: `https://www.dropbox.com/scl/fi/hhn7lththr7kv8ueuygf9/CardiCat_ICLR25.zip?rlkey=9zrk8jasretwyjy4wz1i54xlg&dl=0`.

Additionally, the datasets can be downloaded here: `https://www.dropbox.com/scl/fi/clonvo55gv1llf9sj9i7o/CardiCat_datasets.zip?rlkey=d8hsypmjf79lycfjbdon282rl&dl=0`.

### 3.2 EVALUATION METRICS

Some of the synthetic data quality evaluation metrics were adapted from Xu et al. (2019). Here we elaborate on the specifics of each evaluation metric.

**Marginal reconstruction**.

- Continuous features were evaluated using the complement of the Kolmogorov-Smirnov (KS) statistic, $1 - KS_{F_n, \hat{F}_m}(x_j) = 1 - \sup_x |F_n(x) - \hat{F}_m(x)|$, where $F_n, \hat{F}_m$ are the observed and generated empirical distribution functions, respectively.

- Categorical features were evaluated using the complement of the Total Variation Distance (TVD), $1 - TVD_{R,S} = 1 - \frac{1}{2} \sum_{\ell=1}^{c_j} |R_\ell - S_\ell|$, where $R, S$ are the observed and generated marginal probability measures, respectively .

**Bi-variate reconstruction**.

- Numerical pairs. The complement of the correlation difference between two numerical features $\mathbf{x}_j, \mathbf{x}'_j$ is used for evaluating numerical bi-variate reconstruction, $1 - \frac{|Corr(\mathbf{x}_j, \mathbf{x}_{j'}) - Corr(\hat{\mathbf{x}}_j, \hat{\mathbf{x}}_{j'})|}{2}$.

- Categorical pairs. The complement of the TVD on the contingency table between two categorical features is used for categorical bi-variate reconstruction, $1 - \frac{1}{2} \sum_{\ell=1}^{c_j} \sum_{\ell'=1}^{c_{j'}} |R_{\ell, \ell'} - S_{\ell, \ell'}|$.

- Mixed pairs. Mixed feature pairs were evaluated by averaging reconstruction accuracy of the conditional distributions for the numerical variable $\mathbf{x}_{j'}$ given values for the categorical variable $\mathbf{x}_j$

$$1 - \sum_{\ell=1}^{c_j} \pi_\ell \cdot \sup_{x_{j'}} |F_n(x_{j'}|x_j = \ell) - \hat{F}_m(x_{j'}|x_j = \ell)|,$$

where $F_n, \hat{F}_m$ are the observed and generated empirical distribution functions, respectively, and $\pi_\ell = P(x_j = \ell)$ under the true distribution. If value $\ell$ is unobserved for variable $\mathbf{x}_j$ in the generated, we set the KS for this value to 1.

## 3.3 NETWORK DESIGN AND HYPER-PARAMETERS

**Network design** All benchmark models share the same hidden-layer structure of three 128-128-128 fully connected layers in both the encoder and decoder using ReLu activation functions. CardiCat, tVAE and VAE (vanilla) have a multivariate normal Gaussian prior. In all cases, including tGAN, the size of the networks' latent dimension is set to 15. In terms of data preprocessing, CardiCat and VAE apply label encoding to categorical features, and a shift-scale normalization into a distribution centered around zero with standard deviation of one to numerical variables. One-hot encoding and categorical embeddings are applied according to the main paper. The preprocessing of tGAN and tVAE is done as part of their code library and as specified in Xu et al. (2019).

**Hyperparameters** All models were trained on a train/test split of 80/20 of the dataset. Training was done with 150 epochs, batch sizes of 2,000 and an Adam optimizer with a learning rate of 0.0005 on the train set. The loss factor of the ELBO of all VAEs was set to 5.

## 3.4 CONDITIONAL GENERATOR RESULTS

Table 2: Evaluation results of conditional generators.

| dataset | model | marginal | | pairs | | |
| | | categorical | numerical | categorical | mixed | correlation |
|---|---|---|---|---|---|---|
| Bank | cCardiCatMask | 0.86 | 0.78 | **0.76** | 0.74 | **0.97** |
| | tGAN | **0.88** | **0.87** | 0.59 | **0.82** | 0.95 |
| Census | cCardiCatMask | 0.79 | **0.73** | 0.64 | 0.68 | **0.98** |
| | tGAN | **0.88** | 0.71 | **0.68** | **0.8** | 0.97 |
| Credit | cCardiCatMask | **0.92** | 0.83 | **0.93** | **0.84** | **0.96** |
| | tGAN | 0.87 | **0.86** | 0.64 | 0.83 | 0.91 |
| Criteo | cCardiCatMask | 0.65 | 0.80 | **0.44** | 0.70 | **0.97** |
| | tGAN | **0.78** | **0.86** | 0.36 | **0.73** | 0.95 |
| MIMIC | cCardiCatMask | **0.82** | 0.85 | **0.68** | **0.79** | **0.99** |
| | tGAN | 0.72 | 0.85 | 0.58 | 0.73 | 0.97 |
| Medical | cCardiCatMask | **0.58** | 0.80 | **0.17** | **0.60** | **0.96** |
| | tGAN | 0.58 | **0.90** | 0.06 | 0.59 | 0.94 |
| PetFinder | cCardiCatMask | **0.88** | 0.76 | **0.78** | **0.91** | **0.98** |
| | tGAN | 0.87 | **0.77** | 0.70 | 0.80 | 0.97 |
| Simulated | cCardiCatMask | 0.75 | **0.90** | **0.60** | **0.90** | **0.99** |
| | tGAN | **0.78** | 0.84 | 0.46 | 0.79 | 0.98 |

## 4  RELATED MODELS & LITERATURE

Relevant related models & literature is summarized in table 3

Table 3: Summary of relevant literature related to deep tabular generative models

| Model | architecture | use-case |
|---|---|---|
| RVAE Akrami et al. (2020; 2022) | VAE: two-component mixture likelihoods | Outlier robust |
| HI-VAE Nazabal et al. (2020) | VAE: type specific likelihoods with hierarchical structure | Imputation |
| VAEM Ma et al. (2020) | VAE: hierarchical two-stage structure | Imputation |
| VSAE Gong et al. (2021) | VAE: modeling using imputation mask | Imputation |
| medGAN Choi et al. (2017) | GAN: minibatch averaging, batch norm. | Synthetic patient records |
| table-GAN Park et al. (2018) | GAN: balance between privacy level and model compatibility | Private data synthesis |
| TGAN/CTGAN Xu & Veeramachaneni (2018); Xu et al. (2019) | GAN/VAE: mode-specific norm., conditional generator | Conditional data synthesis |
| CTAB-GAN Zhao et al. (2021) | GAN: conditionally encoding imbalanced mixed type | Private conditional data synthesis |

### 4.1  ADDITIONAL NOTES ON MODELS

**VAEM**. Because the VAEM package available on Github[1] is no longer supported by its dependencies, we wrote our own version of VAEM that fits our benchmark settings. The input to the marginal VAEs are either one-hot encoded categorical variables, or normalized numerical variables. The latent variable has a single dimension, and the output of the decoder is either size one or the one-hot vector size for numerical and categorical features respectively. However, this model performed very poorly on all the datasets, and we have decided not to include it as a benchmark model.

**tGAN**. `tVAE` is used as a benchmark for `tGAN`, a conditional generative adversarial network framework with the same data normalization. `tGAN`'s approach to overcome the imbalance nature of the data is done by "training-by-sampling". Sampled data from their conditional generator aims to represent more accurately the underlying marginal distributions of the categorical features. The conditional generator in `tGAN` is a concatenation of all the one-hot encoded categorical features, where all the elements in the vector are masked (set to zero), except the one-hot elements of the conditional value. During training, the conditional variable for each row is chosen uniformly from the set of all categorical features. A cross entropy term between the conditional value and the respective generated value is added to enforce the conditional generator during training. In contrast to `tGAN`, `cCardiCatMask`, does not employ such a "training-by-sampling" nor an additional cross entropy term between the conditional value and the respective generated value.

---

[1]https://github.com/microsoft/VAEM