# OpenReview forum: "CardiCat: a Variational Autoencoder for High-Cardinality Tabular Data"
_ICLR.cc/2025/Conference — Submitted to ICLR 2025_

### Official Review · Reviewer_xrEX · 2024-11-03

**Soundness:** 3
**Presentation:** 4
**Contribution:** 2
**Rating:** 3
**Confidence:** 4

**Summary:**

The authors focus on the design of a better variational autoencoder architecture and training objective for tabular data, CardiCat. Specifically, they focus on the challenging task of modeling high cardinality categorical features with class imbalance. The standard approach with one-hot encodings is very expensive, introducing many more parameters and therefore increasing sample complexity. The core technique of this paper is to substitute one-hot encoding with a low-dimensional embedding layer used in both the encoder and decoder. Then, the reconstruction loss for these features is computed with MSE in embedding space instead of using a cross-entropy loss in the raw label space. To prevent embedding collapse, the authors propose a variance-based regularization term. Small-scale evaluations demonstrate that CardiCat outperforms vanilla VAE baselines in recovering marginal and pairwise conditional distributions on a variety of tabular datasets.

**Strengths:**

* Well-motivated: The problem setting is important and underappreciated (VAEs for modeling heterogenous tabular datasets), so this work is well-motivated.
* Clear writing: The paper is well-written and contextualized well in the VAE literature.
* Correctness: All mathematical statements appear correct.
* Appears reproducible: The method is presented in enough detail that I believe it could be reproduced straightforwardly.

**Weaknesses:**

*Unconvincing evaluations.*

My major concern is that (1) the CardiCat framework gives up on optimizing a variational lower bound of the log-likelihood, which makes model comparison far more challenging, and moreover (2) does not provide convincing surrogate evaluations for sample quality or diversity.

In particular, the CardiCat objective (L228) is no longer a valid ELBO, and therefore it is not possible for the authors to directly compare the ELBO of their model versus other VAEs. (Aside: this fact was obfuscated by the writing near L218-220, where it appeared that the objective is indeed a valid ELBO. I would urge the authors to edit this writing to make clear that the CardiCat objective is not an ELBO.)

It is OK to not use a likelihood-based model as long as the downstream evaluations of sample quality and diversity are convincing. My concern is that they are not. The authors report two main metrics: matching the marginals of each feature distribution and pairwise conditionals between features. I did not find this to be a realistic test of the sample quality and diversity of their model. I recognize that evaluation of non-likelihood-based generative models for tabular data is challenging (there are no standard metrics like FID), but I would have at least hoped the authors could test the quality of their learned representations for supervised tasks. The experiments leave me unconvinced that CardiCat actually models the joint distribution $p(x)$ better than the alternatives.

In addition, the evaluation is done on a very small scale. The authors mention that they intentionally use a simple setting for a more direct comparison to VAE---I was not very convinced by this. Ideally, one would compare directly against state-of-the-art methods at a reasonably large scale, perhaps using their architectures etc. and showing that the new contribution (the CardiCat dual embedding and regularizer) improves performance.

*Giving up ELBO seems unnecessary.*

I am not convinced that it is even necessary to give up the ELBO in order to avoid one-hot embeddings. For example, Plaid [1] is a diffusion language model which uses low-dimensional embeddings for categorical data and still preserves the ELBO objective, allowing direct model comparison with autoregressive and other generative models.

*Lack of motivation for regularization term.*
The embedding regularization term (L245) was a little surprising: it regularizes the sum of the variances? Why not compute the element-wise variances as $V_j({e_j})$ and $V_j({e^0_j})$ then have the regularization term be $||V_j({e_j}) - V_j({e^0_j})||^2_2$? I believe readers would appreciate some more motivation for this.

[1] Ishaan Gulrajani, Tatsunori B. Hashimoto. Likelihood-Based Diffusion Language Models. In NeurIPS, 2023. https://proceedings.neurips.cc/paper_files/paper/2023/hash/35b5c175e139bff5f22a5361270fce87-Abstract-Conference.html

**Questions:**

I would be willing to improve my score if the authors could:
1. Demonstrate clearly why a likelihood-based objective is not reasonable, thereby justifying their non-likelihood-based objective which sacrifices direct model comparison with the ELBO.
2. Improve the writing around the objective (L218-220) to make clear that the CardiCat objective is not a bound of the likelihood (mentioned in Weaknesses above).
3. Expand the scope of their evaluations to larger-scale settings with other tests of model quality, such as joint sample quality / diversity or supervised probes on representations.

---

> ### Author Response · Authors · 2024-11-24
> **Answering Reviewer xrEX**
>
> Thank you for taking the time to review our paper. We appreciate your feedback, which has greatly contributed to the enhancement of our work. Your comments have been invaluable in refining the content and strengthening the overall quality of the paper.
>
> That’s correct: the deviation from a traditional ELBO formulation is due to (a) the categorical embeddings parameterization, and (b) the regularization term in the loss. Our categorical embeddings are treated in the loss as the the other numerical features, which in effect do not translate into probabilities directly. This in return deviates from a purely generative probabilistic model in behalf of ”pseudo”-likelihoods. We believe that the traditional ELBO can be still preserved by normalizing the embedding space for each categorical feature (e.g., using a softmax function) so the embedding loss component represent the mean of the normalized deviations from the ”true” embedding (similarly to Gulrajani & Hashimoto (2024)).
>
> Our framework is implemented on a simple VAE architecture in order to cleanly demonstrate the advantages of such embedding framework. Recent diffusion-based models have been showing competitive results against the more conventional VAE and GAN methods (Kotelnikov et al., 2023; Zhang et al., 2023; van Breugel et al., 2024). We believe these recent models do not take away from CardiCat’s contribution because their contribution is parallel. For example, CardiCat can be applied to latent-space diffusion-based models as well (such as in the architecture used by Zhang et al. (2023)), and in combination of other competing models (such as in (Xu et al., 2019; Stoian et al., 2024). Additionally, many of these publications do not assert one model to be a clear winner in all evaluation metrics, with the more ”conventional” approaches still performing well in some (van Breugel et al., 2024).
>
> There are two substantial difficulties with learning distributions with high cardinality variables: a. resampling from the full distribution, including rare categorical values, and b. learning the dependencies between categorical values and other variables. We show substantial gains in both of these elements by looking at marginal and bi-variate distributions. Related works usually include low-order statistics evaluation metrics (by including various marginal and bi-variate density estimation and diversity metrics) and machine learning efficiency metrics (by evaluating the synthetic data against the real data using a classification/regression task). We chose to focus on low order statistics and provide an additional ’mixed’ bi-variate evaluation metric to emphasize the need to assess also mixed-type interactions. We also demonstrated that concentrating on one aggregative score that averages the evaluation metric can be misleading: our results show that different models excel in different metrics. Machine learning efficiency metrics can be misleading as well because they concentrate on a discriminant function and not on the joint distribution efficacy of the generative model.
>
> Once again, thank you for your valuable feedback. Sincerely.

---

> ### Comment · Reviewer_xrEX · 2024-11-25
> **Response to Author Rebuttal**
>
> Thanks very much for your response.
>
> Unfortunately my main concern that the baselines are outdated was not resolved, and it is not clear to me that the proposed method would outperform modern models for tabular data (such as the diffusion or GAN approaches you cite). You suggest that you have chosen specifically to demonstrate these changes on "a simple VAE architecture", which I did not find convincing. Vanilla VAEs are known to have plenty of issues, such as posterior collapse and poor sample quality, and therefore it does not feel reasonable to claim that improving over them is an empirical contribution---you should compare to modern baselines and beat them. For example, even if one proposes an improvement to MLPs, they should demonstrate it in some setting of practical interest, such as plugging their method into a modern Transformer architecture and showing reasonable results.
>
> Because this issue is unresolved, I will keep my score unchanged.

---

### Official Review · Reviewer_AQFr · 2024-11-03

**Soundness:** 1
**Presentation:** 3
**Contribution:** 1
**Rating:** 3
**Confidence:** 4

**Summary:**

The paper proposes CardiCat, a VAE-based generative model designed to handle high-cardinality categorical features in tabular data by using dual encoder-decoder embedding layers. The authors claim that this approach avoids the need for one-hot encoding, reduces the number of trainable parameters, and provides a compact parameterization that improves the model’s ability to capture complex dependencies. Empirical results indicate that CardiCat outperforms traditional VAE models and other baselines.

**Strengths:**

- The method addresses a well-known and relevant problem.
- The structure of the paper is well-organized.

**Weaknesses:**

- The contribution appears technically minimal or lacks sufficient justification.
- Certain theoretical aspects require further review and clarification.
- The related work section provides only a high-level overview and omits several relevant references.
- Additional baselines are needed to strengthen the empirical evidence supporting the contributions and demonstrate their significance.

**Questions:**

### Theoretical Issues

- The relevance of the proposed method is not entirely clear. VAEM individual VAEs already preprocess input data into a smooth latent space, and the posterior of the uni-dimensional VAEs can be seen as an embedding similar to CardiCat. Thanks to this, VAEM’s dependency model learns the inter-feature dependencies effectively. Notably, VAEM generalizes CardiCat by using a VAE framework, while CardiCat employs a simpler, deterministic autoencoder (AE) with regularization applied through a loss term. Why would CardiCat theoretically outperform VAEM? Line 160 references Section 4 for empirical justification, but this evidence is not provided.

- The authors claim that "This allows us to avoid altogether the need to one-hot encode the non-binary categorical features at any point in the process." However, the significance of avoiding one-hot encoding is unclear. With an appropriate design, VAEM, and possibly more recent methods for heterogeneous data [2-5], could potentially achieve comparable results while maintaining a similar parameter count as CardiCat.

- While the model is framed as a VAE adaptation, the reconstruction loss relies on mean-squared errors and cross-entropy rather than likelihoods, with added regularization terms. These modifications make the objective to diverge from the traditional ELBO used in VAEs. Although this optimization approach may still be effective (since the Gaussian pdf’s exponent is effectively a squared error weighted by variance), it deviates from a purely generative probabilistic model.

- In line 258, the likelihood is defined as a factorized Gaussian, which conflicts with the loss function described in Section 3.3.

- It is unclear how the decoded embeddings of categorical features are transformed back into parameters for categorical distributions.

- One of the strengths of VAEs is their ability to approximate likelihoods. How can likelihood approximations be evaluated within the proposed model?

- Technical inaccuracies:
  - Line 256: "parametrization" should be replaced with "reparameterization" [1].

### Experimental Issues

#### Baselines

- The baselines employed are insufficient and outdated, with the most recent comparison model dating back to 2020. Including the original VAE from 2014, which has limited value given the significant advancements in handling heterogeneous data, undermines the comparison. Why were recent methods [2-5] for heterogeneous missing data not considered as baselines? The comparison with tGAN is not adequately discussed in the text, and its relevance remains unclear.

### Minor Comments

- Typographical errors:
  - Quotation mark issues (e.g., lines 49 and 288).

[1] Kingma, Diederik P., and Max Welling. "Auto-encoding variational bayes." arXiv preprint arXiv:1312.6114 (2013).

[2] Ma, Chao, et al. "VAEM: a deep generative model for heterogeneous mixed type data." Advances in Neural Information Processing Systems 33 (2020): 11237-11247.

[2] Peis, Ignacio, Chao Ma, and José Miguel Hernández-Lobato. "Missing data imputation and acquisition with deep hierarchical models and Hamiltonian Monte Carlo." Advances in Neural Information Processing Systems 35 (2022): 35839-35851.

[3] Antelmi, Luigi, et al. "Sparse multi-channel variational autoencoder for the joint analysis of heterogeneous data." International Conference on Machine Learning. PMLR, 2019.

[4] Gong, Yu, et al. "Variational selective autoencoder: Learning from partially-observed heterogeneous data." International Conference on Artificial Intelligence and Statistics. PMLR, 2021.

---

> ### Author Response · Authors · 2024-11-24
> **Answering Reviewer AQFr**
>
> Thank you for taking the time to review our paper. We appreciate your feedback, which has greatly contributed to the enhancement of our work. Your comments have been invaluable in refining the content and strengthening the overall quality of the paper.
>
> ANSWER 1 (The relevance of the proposed method):
>
> Ma et al. (2020) is based on a two- stage structure where the first stage individual features are homogenized by learning an independent VAE for each feature. Then, the separately learned factorized latent variables from the first stage models are used as inputs to a second stage VAE. The first stage independent models are trained to learn the marginal distribution of each feature. The second stage model is expected to ”recover” the joint distribution, using only information from the marginal distributions. However, if the features are not independent, there is much more information in a joint distribution than can be captured by its marginal distributions. We tested that empirically, but unfortunately, because the authors’ original code-base is not functional
> we were not able to use their code directly and had to test their framework using our own implementation of their model. Because of that, and the subsequent subpar performance, we didn’t include the results of our own implementation.
>
> ANSWER 2 ( the objective to diverge from the traditional ELBO used in VAEs):
>
> That’s correct: the deviation from a traditional ELBO formulation is due to (a) the categorical embeddings parameterization, and (b) the regularization term in the loss. Our categorical embeddings are treated in the loss as the the other numerical features, which in effect do not translate into probabilities directly. This in return deviates from a purely generative probabilistic model in behalf of ”pseudo”-likelihoods. We believe that the traditional ELBO can be still preserved by normalizing the embedding space for each categorical feature (e.g., using a softmax function) so the embedding loss component represent the mean of the normalized deviations from the ”true” embedding (similarly to Gulrajani & Hashimoto (2024)).
>
>
> ANSWER 3 ( The baselines employed are insufficient and outdated):
>
> Our framework is implemented on a simple VAE architecture in order to cleanly demonstrate the advantages of such embedding framework. Recent diffusion-based models have been showing competitive results against the more conventional VAE and GAN methods (Kotelnikov et al., 2023; Zhang et al., 2023; van Breugel et al., 2024). We believe these recent models do not take away from CardiCat’s contribution because their contribution is parallel. For example, CardiCat can be applied to latent-space diffusion-based models as well (such as in the architecture used by Zhang et al. (2023)), and in combination of other competing models (such as in (Xu et al., 2019; Stoian et al., 2024). Additionally, many of these publications do not assert one model to be a clear winner in all evaluation metrics, with the more ”conventional” approaches still performing well in some (van Breugel et al., 2024).
>
>
> Once again, thank you for your valuable feedback. Sincerely.

---

> ### Comment · Reviewer_AQFr · 2024-11-26
> **Rebuttal Response**
>
> Thank you to the authors for their responses. However, most of my concerns, some of which are shared with Reviewer xrEX, remain unaddressed. Specifically:
>
> - Unless I am misunderstanding something, in the VAEM approach, when the data dimensions are not independent, their corresponding marginal posteriors should preserve these interdependencies. This technique is designed as a more balanced approach to effectively capture such relationships, making it a very robust alternative.
>
> - The related work section is outdated. While this can be resolved by rewriting the literature review, including more recent baseline methods is essential to justify the significance of this paper. Comparing CardiCat to older methods does not seem fair or sufficient.
>
> - The objective deviates from the ELBO, not only due to how categorical data is processed but also because the reconstruction term differs from the conditional likelihood. To properly denote an objective as an ELBO, it should ideally form a lower bound on the log-evidence:
>   $$
>   \mathbb{E}_{q(z|x)}[\log p(x|z)] - KL(q(z)||p(z)) \leq \log p(x)
>   $$
>
> For these reasons, I will keep my score unchanged.

---

### Official Review · Reviewer_uHsb · 2024-11-03

**Soundness:** 3
**Presentation:** 3
**Contribution:** 2
**Rating:** 5
**Confidence:** 3

**Summary:**

In this paper, authors have proposed a new method called CardiCat that can accurately fit the imbalanced high-cardinality and heterogeneous tabular data. It employs a dual encoder-decoder embedding layers architecture and a customized loss function that computes the reconstruction in the embedding space. The model was tested on 8 datasets and showed a better performance compared to other methods.

**Strengths:**

The authors introduced a novel to fit the imbalanced tabular data. The paper is easy to follow and understand.

The results in the Table 2 shows better performance than other VAE based methods.

**Weaknesses:**

Lack of state-of-the-art comparative methods. Most of the comparative methods are methods before (vae, tvae) 2019, while the most advanced methods are necessary.

In Figure 3, the proposed model seems to have similar or worse performance than tGAN, especially for the marginal reconstruction. In Table 2, do you have any comparisons with tGAN?

**Questions:**

The datasets do not seem to be high cardinality. What if the number of cardinalities is greater than the number of samples?

The evaluation metrics seem to be limited. Are there any experiments showing the generated data's performance?

---

> ### Author Response · Authors · 2024-11-24
> **Answering Reviewer uHsb**
>
> Thank you for taking the time to review our paper. We appreciate your feedback, which has greatly contributed to the enhancement of our work. Your comments have been invaluable in refining the content and strengthening the overall quality of the paper.
>
> ANSWER 1 (Lack of state-of-the-art comparative methods):
>
> Our framework is implemented on a simple VAE architecture in order to cleanly demonstrate the advantages of such embedding framework. Recent diffusion-based models have been showing competitive results against the more conventional VAE and GAN methods (Kotelnikov et al., 2023; Zhang et al., 2023; van Breugel et al., 2024). We believe these recent models do not take away from CardiCat’s contribution because their contribution is parallel. For example, CardiCat can be applied to latent-space diffusion-based models as well (such as in the architecture used by Zhang et al. (2023)), and in combination of other competing models (such as in (Xu et al., 2019; Stoian et al., 2024). Additionally, many of these publications do not assert one model to be a clear winner in all evaluation metrics, with the more ”conventional” approaches still performing well in some (van Breugel et al., 2024).
>
>
> ANSWER 2 ( similar or worse performance than tGAN):
>
> The results of the conditional generators can be found in the supplementary material (the table in page 3). While the marginal reconstruction scores are indeed a bit below the tGAN scores, the bi-variate scores all exceed those of tGAN’s. In addition, tGAN employs a ”training by sampling” scheme during training, which can artificially boost the marginal reconstruction of categorical features  (where, the conditional and training data are sampled according to the log-frequency of each category). It is evident from the bi-variate
> metrics that CardiCat was able to learn better the joint-distribution between features.
>
> ANSWER 3 (What if the number of cardinalities is greater than the number of samples?):
>
> If the cardinality of a categorical feature is equal to or almost the number of samples of the data, than this feature is degenerate in the sense that it acts more of an index than a feature that adds a valuable signal. Unlike many other in the literature, we demonstrate our model also on non-trivial datasets in terms of cardinality. This type of high cardinality is often overlooked by many tabular generative models, and we believe our model makes a concrete contribution towards modeling tabular mixed-type data. There is an interest in exploring the cases where the total cardinality of the data/feature is very high relative to the number of samples, but in this paper we don’t focus on solving these type of scenarios.
>
> ANSWER 4 (The evaluation metrics seem to be limited):
>
> There are two substantial difficulties with learning distributions with high cardinality variables: a. resampling from the full distribution, including rare categorical values, and b. learning the dependencies between categorical values and other variables. We show substantial gains in both of these elements by looking at marginal and bi-variate distributions. Related works usually include low-order statistics evaluation metrics (by including various marginal and bi-variate density estimation and diversity metrics) and machine learning efficiency metrics (by evaluating the synthetic data against the real data using a classification/regression task). We chose to focus on low order statistics and provide an additional ’mixed’ bi-variate evaluation metric to emphasize the need to assess also mixed-type interactions. We also demonstrated that concentrating on one aggregative score that averages the evaluation metric can be misleading: our results show that different models excel in different metrics. Machine learning efficiency metrics can be misleading as well because they concentrate on a discriminant function and not on the joint distribution efficacy of the generative model.
>
>
> Once again, thank you for your valuable feedback. Sincerely.

---

### Official Review · Reviewer_JzTs · 2024-11-07

**Soundness:** 1
**Presentation:** 2
**Contribution:** 1
**Rating:** 5
**Confidence:** 4

**Summary:**

This paper aims to address the generation of high-cardinality tabular data by proposing CardiCat, a variational autoencoder (VAE) model that employs regularised dual encoder-decoder embedding layers.

**Strengths:**

1. The paper is generally well-written.
2. The authors follow consistent notations throughout the paper.
3. The code is provided.

**Weaknesses:**

**1. [Important] Seemingly inaccurate claim of contribution.** CardiCat does not seem to be the first to employ dual embeddings in tabular data generation. I would suggest the authors refer to some recent papers, like TabSyn [1], where the VAE is equipped with a trainable tokeniser as 1. CardiCat.

**2. [Important] Incomprehensive comparison to benchmark methods.** The paper seems to only include some conventional VAE and GAN methods for comparison. However, there has been some recent work on generating tabular data with mixed types [1]. I would suggest the authors refer to them and at least include some of the recent methods for a more general comparison.

**3. [Important] Evaluation metrics are not comprehensive.** Following the above concern on benchmark methods. Usually, it would be insufficient and inconclusive to only evaluate the generator with marginal and bi-variate statistical fidelity metrics. Please refer to the literature [2, 3, 4] for more indicative metrics like downstream performance and multivariate fidelity metrics.

**4. [Important] Unclear descriptions of conditional CardiCat.** And the corresponding results of conditional CardiCat seem missing in the paper.

**5. Code is a bit hard to go through.** I carefully checked the provided codebase. Although it is not necessary to have clear-to-read code for everyone, the current open-source version seems somewhat messy. One example is that the comments for functions remain unfinished: get_pred in src/postprocessing.py, the explanations of arguments are simply `_description_`. I would suggest the authors clean their codebase to save time for potential users.


[1] Zhang, Hengrui, et al. "Mixed-type tabular data synthesis with score-based diffusion in latent space." arXiv preprint arXiv:2310.09656 (2023).

[2] Stoian, Mihaela C., et al. "How Realistic Is Your Synthetic Data? Constraining Deep Generative Models for Tabular Data." The Twelfth International Conference on Learning Representations.

[3] Ma, Junwei, et al. "TabPFGen--Tabular Data Generation with TabPFN." arXiv preprint arXiv:2406.05216 (2024).

[4] Qian, Zhaozhi, Bogdan-Constantin Cebere, and Mihaela van der Schaar. "Synthcity: facilitating innovative use cases of synthetic data in different data modalities." arXiv preprint arXiv:2301.07573 (2023).

**Questions:**

1. Why not treat binary features as categorical features?

---

> ### Author Response · Authors · 2024-11-24
> **Answering Reviewer JzTs**
>
> Thank you for taking the time to review our paper. We appreciate your feedback, which has greatly
> contributed to the enhancement of our work. Your comments have been invaluable in refining the content
> and strengthening the overall quality of the paper.
>
>
> ANSWER 1 (Seemingly inaccurate claim of contribution):
>
> Indeed, TabSyn ( (Zhang et al., 2023)) utilizes a feature tokenizer that converts each column (both numerical and categorical) into a d-dimensional vector. However, they also apply a detokenizer to the recovered token representations in the encoder. There are three main key differences between CardiCat’s embeddings and TabSyn’s tokenizer that makes CardiCat’s embeddings superior.
> (1) CardiCat’s loss calculation is done directly on the embedding space while TabSyn is done on the original features. (2) The dual-embeddings used in CardiCat is trained to have the same embedding (i.e., weights) between the encoder and decoder (hence the ”dual”), while TabSyn enforces no such parity ending up with two different tokenizers for each feature. (3) CardiCat’s avoids one-hot encoding the categorical features while TabSyn tokenizer expect one-hot encoded categorical features, which can be very expensive in terms of the model’s required learned parameters ( especially for categorical features with high cardinality).
>
> ANSWER 2 (incomprehensive comparison to benchmark methods):
>
> Our framework is implemented on a simple VAE architecture in order to cleanly demonstrate the advantages of such embedding framework. Recent diffusion-based models have been showing competitive results against the more conventional VAE and GAN methods (Kotelnikov et al., 2023; Zhang et al., 2023; van Breugel et al., 2024). We believe these recent models do not take away from
> CardiCat’s contribution because their contribution is parallel. For example, CardiCat can be applied to latent-space diffusion-based models as well (such as in the architecture used by Zhang et al. (2023)), and in combination of other competing models (such as in (Xu et al., 2019; Stoian et al., 2024). Additionally, many of these publications do not assert one model to be a clear winner in all evaluation metrics, with the more ”conventional” approaches still performing well in some (van Breugel et al., 2024).
>
> ANSWER 3 (valuation metrics are not comprehensive):
>
>  There are two substantial difficulties with learning distributions with high cardinality variables: a. resampling from the full distribution, including rare categorical values, and b. learning the dependencies between categorical values and other variables. We show substantial gains in both of these elements by looking at marginal and bi-variate distributions. Related works usually include low-order
> statistics evaluation metrics (by including various marginal and bi-variate density estimation and diversity metrics) and machine learning efficiency metrics (by evaluating the synthetic data against the real data using a classification/regression task). We chose to focus on low order statistics and provide an additional ’mixed’ bi-variate evaluation metric to emphasize the need to assess also mixed-type interactions. We also demonstrated that concentrating on one aggregative score that averages the evaluation metric can be misleading: our results show that different models excel in different metrics. Machine learning efficiency metrics can be misleading as well because they concentrate on a discriminant function and not on the joint distribution efficacy of the generative model.
>
> ANSWER 4 (unclear descriptions of conditional CardiCat):
>
> The results of the conditional generators can be found in the supplementary material (page 3).
>
> ANSWER 5 (Code is a bit hard to go through):
>
> Thank you for bringing this under our attention. This specific issue was a result of an unnoticed failed doc-string compiler, which is fixed now. As oppose to many published open-source code-bases, we are proud to adhere to pep8 standards in addition to provide rigorous documentation
>
> Questions: Why not treat binary features as categorical features?
>
> In therms of the loss, the embedded categorical relationship between the binary values will stay fixed regardless of the resulting embedding map.
>
>
> Once again, thank you for your valuable feedback.
> Sincerely.

---

> ### Comment · Reviewer_JzTs · 2024-11-28
> **Thank the authors for their response**
>
> I would like to thank the authors for their feedback on my comments. And I acknowledge that their response resolves some of my concerns about the paper. Below are the remaining ones:
>
> 1. If CardiCat can work as a pluggable module of any VAE-based models, I would suggest the authors to support the claim with empirical evidence. Otherwise, it is not sufficiently convincing (also noted by Reviewer xrEX).
>
> 2. I am unsure if I fully agree with the authors on their feedback for evaluation metrics. I agree with the authors that downstream accuracy is not the optimal metric to demonstrate how well the generative model captures the joint distribution. However, it is still complementary to the low-order metrics, as many downstream models need to attend to higher-order relationships for better generation quality. Therefore, the downstream accuracy can empirically serve as a good "proxy" for the generators' utility. And "imperfection" may not be convincing enough to justify why the authors do not evaluate the downstream performance.
>
> 3. The authors' feedback on "Why not treat binary features as categorical features?" remains unclear to me. What does "therms of the loss" refer to? Why does "embedded categorical relationship" stay fixed?
>
> Looking forward to the authors' response.

---

### Author Response · Authors · 2024-11-24
**References for comments**

Ishaan Gulrajani and Tatsunori B Hashimoto. Likelihood-based diffusion language models. Advances in
Neural Information Processing Systems, 36, 2024.

Akim Kotelnikov, Dmitry Baranchuk, Ivan Rubachev, and Artem Babenko. Tabddpm: Modelling tabular
data with diffusion models. In International Conference on Machine Learning, pp. 17564–17579.
PMLR, 2023.

Chao Ma, Sebastian Tschiatschek, Richard Turner, Jos´e Miguel Hern´andez-Lobato, and Cheng Zhang.
Vaem: a deep generative model for heterogeneous mixed type data. Advances in Neural Information
Processing Systems, 33:11237–11247, 2020.

Mihaela C˘at˘alina Stoian, Salijona Dyrmishi, Maxime Cordy, Thomas Lukasiewicz, and Eleonora
Giunchiglia. How realistic is your synthetic data? constraining deep generative models for tabular
data. arXiv preprint arXiv:2402.04823, 2024.

Boris van Breugel, Jonathan Crabb´e, Rob Davis, and Mihaela van der Schaar. Latable: Towards large
tabular models. arXiv preprint arXiv:2406.17673, 2024.

Lei Xu, Maria Skoularidou, Alfredo Cuesta-Infante, and Kalyan Veeramachaneni. Modeling tabular data
using conditional gan. Advances in Neural Information Processing Systems, 32, 2019.

Hengrui Zhang, Jiani Zhang, Balasubramaniam Srinivasan, Zhengyuan Shen, Xiao Qin, Christos Falout-
sos, Huzefa Rangwala, and George Karypis. Mixed-type tabular data synthesis with score-based
diffusion in latent space. arXiv preprint arXiv:2310.09656, 2023

---

### Comment · Area_Chair_sED8 · 2024-11-25
**Discussion between reviewers and authors**

Time for discussions as author feedback is in. I encourage all the reviewers to reply. You should treat the paper that you're reviewing in the same way as you'd like your submission to be treated :)

---

### Meta-Review · Area_Chair_sED8 · 2024-12-20

**Metareview:**

This paper tackles tabular synthetic data generation in the situation of mixed-type data and high cardinality of the categorial variables. The two main contributions seem to be (1) a modified VAE training objective by adding a regularisation term on the feature embeddings, and (2) an encoder-decoder architecture for the feature embeddings. Experiments on synthetic data generation (evaluated by cardinality feature reconstruction and data statistics) shows promising results.

Reviewers have major issues with this paper:

1. Unclear presentation in terms of positioning the novelty of the work in related literature, e.g., the innovation regarding the feature embedding, and justification for the objective deviating from the VAE loss.

2. Lack of other state-of-the-art baselines in comparison.

The author feedback did not address well the reviewers' concerns.

PS - a concern of author identity leakage was raised by a reviewer regarding the provided code.

**Additional Comments On Reviewer Discussion:**

Reviewer - author discussions: author rebuttal did not change the mind of most of the reviewers, major concerns not addressed.

---

### Decision · Program_Chairs · 2025-01-22

Reject